# Reading Equality into Asymmetry: Dual Ordination in the Eyes of Modern Chinese *Bhikṣuṇī*s

Ester Bianchi 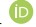

Department of Philosophy, Social Sciences and Education, University of Perugia, 06123 Perugia, Italy; ester.bianchi@unipg.it

**Abstract:** The "Dual Ordination" (*erbuseng jie* 二部僧戒) is a *Vinaya*-based ordination procedure introduced to China from Śrī Laṅkā in the fifth century; in the late imperial period it came to be included in the main ordination system. It stipulates that full ordination for nuns is to be carried out first in front of an assembly of *bhikṣuṇī*s and then another assembly of *bhikṣu*s. However, contrary to this stipulation, ordinations have mainly been conferred to women by *bhikṣu*s alone in China since the tenth century. The Dual Ordination procedures became a topic of discussion during the Republic of China (1911–1949) with the result that it was eventually reintroduced on the Mainland at the beginning of the 1980s, mainly due to the efforts of *bhikṣuṇī*s Longlian 隆蓮 (1909–2006) and Tongyuan 通願 (1913–1991). The article traces the roots of the restoration of Dual Ordinations during the Republican era and provides an account of their history since the 1980s. Finally, Longlian's views about *bhikṣuṇī* ordination are discussed. The objective is to probe the historical and ideological context for the reestablishment of this ordination system in modern and contemporary China, which ultimately strengthened the role and position of Chinese *bhikṣuṇī*s.

**Keywords:** Dual Ordination; *erbuseng jie* 二部僧戒; *bhikṣuṇī* ordination; Longlian 隆蓮; Tongyuan 通願

A boat of compassion from the heavenly sea of the Land of the Lion comes from far away to set up the Dual Ordination platform. Strictly purifying the *Vinaya*, the jade flute of the Discipline blows away the dust of defilement.

天海慈航 獅子國萬里遠來 建二部戒壇嚴淨毗尼 玉律共調離垢地 (Longlian)

## 1. Introduction

In this couplet, hung in the Tiexiang nunnery 鐵像寺 (Chengdu, Sichuan), Chinese *bhikṣuṇī* Longlian 隆蓮 (1909–2006) describes the event celebrated as the beginning of a proper female monastic lineage in China,[1] i.e., the introduction of the procedures for full ordination from Śrī Laṅkā in the fifth century.[2] The procedure is known as "Dual Ordination" (*erbuseng jie* 二部僧戒), and its origins are traditionally traced back to the very beginnings of the female monastic order (*bhikṣuṇīsaṃgha*) at the time of the Buddha. According to the traditional narrative, Buddha Śākyamuni agreed to the requests of his aunt and foster mother Mahāprajāpatī (Ch. Daaidao 大愛道, which, significantly, is reflected in the name of Longlian's other nunnery, Aidao hall 愛道堂) and admitted women into the monastic order, provided that *bhikṣuṇī*s respected the *gurudharma*s (*ba jingfa* 八敬法), eight rules never to be transgressed. These rules, which have been met with a new surge of interest in modern China,[3] were meant to prevent the disappearance of Buddhism from the world after the creation of the *bhikṣuṇīsaṃgha* and clearly subjugated *bhikṣuṇī*s to the *bhikṣusaṃgha*.[4] In the *Vinaya* of the Dharmaguptaka (*Sifenlü* 四分律: *T* no. 1428), the standard *Vinaya* reference in China since the seventh century, the fourth rule of the *gurudharma*s reads: "After having been trained in the six rules for two years as a probationer (*śikṣamāṇā*), the ordination ceremony of a *bhikṣuṇī* has to be carried out in both *saṃgha*s"

(*T* no. 1428: 923b8–10, tr. Heirman 1997, p. 36). The Dual Ordination procedure thus stipulates that nuns' ordination be divided into two sequential steps, and that it be carried out in succession first in front of an assembly of *bhikṣuṇī*s (fully ordained women) and then of *bhikṣu*s (fully ordained men).[5]

In contemporary China, Dual Ordinations are conferred as part of the "Triple Platform Ordination" (*santan dajie* 三壇大戒), a system which includes the bestowal, in succession and during a unique ordination period, of the ten precepts of the *śrāmaṇera*/*śrāmaṇerikā* (male and female "novices"),[6] of the hundreds of precepts for *bhikṣu*s and *bhikṣuṇī*s,[7] and finally of the Bodhisattva precepts.[8] This system was delineated within the Nanshan *Vinaya* tradition (*Nanshan lü* 南山律), which is notably based on the *Vinaya* of the Dharmaguptaka, as late as the seventeenth century.[9]

As for the dual procedures for *bhikṣuṇī*s, they were introduced from Śrī Laṅkā in the fifth century but mostly discarded after the Song dynasty (960–1279).[10] At the beginning of the Qing dynasty (1644–1911), Shuyu 書玉, a *Vinaya* master belonging to the same lineage as the masters who conceived the Triple Platform Ordination system, authored the "Dual Ordination Procedures" (*Erbuseng shoujie yishi* 二部僧受戒儀式, *X* no. 1134), inscribing this early procedure for *bhikṣuṇī* full ordination within the major ordination system of late imperial China.[11] In recent years, Shuyu's text has become the principal reference for Dual Ordinations in China again. However, throughout the Qing dynasty and during the Republic of China (1911–1949), ordinations continued to be conferred to the *bhikṣuṇī*s by only ten *bhikṣu* masters, and thus in disregard of the dual procedures. It should be noted that, in contrast to other *Vinaya* traditions, the Chinese tradition considers ordinations carried out by *bhikṣu*s alone to be valid, as both Guṇavarman (Qiunabamo 求那跋摩) (367–431), a *Vinaya* master involved in the first Dual Ordination, and Daoxuan 道宣 (596–667), the supposed initiator of the Nanshan *Vinaya* tradition, believed that such ordinations produced a minor offense on the part of the *bhikṣu*s conferring the precepts without invalidating the ordination of the *bhikṣuṇī*s undergoing the procedure.[12]

The eight *gurudharma*s also include a reference to the figure of the *śikṣamāṇā*, the two-year female probationer of whom there exists no male counterpart.[13] *Śikṣamāṇā* ordination is a step eventually leading to full ordination and should be conferred at the minimum age of eighteen. However, as Heirman (2008) has shown, this figure was never common in imperial China. On the other hand, it was discussed and referenced by *Vinaya* masters during the Republican Era, and it has partially been revived since the 1980s both in the People's Republic of China (PRC) and in Taiwan (Heirman and Chiu 2012; Chiu and Heirman 2014).

These rules for *bhikṣuṇī* ordination became a topic of discussion among Buddhist circles during the Republican era, but they were implemented in Taiwan in the 1970s (Li Forthcoming) and in Mainland China only in the 1980s. The latter was mainly due to the efforts of two prominent *bhikṣuṇī*s: the aforementioned Longlian (1909–2006), who was based in Sichuan, and her life-long friend Tongyuan 通願 (1913–1991), from Wutaishan.[14] The first *bhikṣuṇī* ordination after the Cultural Revolution, organized by Longlian and Tongyuan, was conferred according to the Dual Ordination system and was held successively in Chengdu's Tiexiang nunnery and Wenshu temple 文殊院 in 1982. Significantly, the candidates involved were all *śikṣamāṇā*s, as it was also Longlian and Tongyuan's intention to (re-)establish this *Vinaya* figure within contemporary Chinese Buddhism.

After this, Dual Ordinations progressively became the most common procedure for *bhikṣuṇī* ordination in Mainland China. Since the year 2000, state regulations have clearly stipulated that *bhikṣuṇī* ordinations must follow Dual Ordination procedures. Ordinations conferred by *bhikṣu*s alone have thus become illegal in the PRC (Bianchi 2019). As for the *śikṣamāṇā*, even if it has not become a rule, this figure is nowadays less exceptional than before. Since *śrāmaṇerikā* ordination is to be conferred before the *śikṣamāṇā* one, and since the latter involves a probationary period of two years before full ordination, *śikṣamāṇā*s participating in Triple Platform Ordinations retake the *śrāmaṇerikā* precepts as a renewal.

In all cases (a woman living in a nunnery without any formal ordination, a *śrāmaṇerikā* or a *śikṣamāṇā*), the time a woman spends in a nunnery between tonsure and full ordination is meant to serve as a training period.[15] Finally, today, *śrāmaṇerikā* ordination is mostly bestowed by an *upādhyāyinī* (female master of the discipline),[16] rather than by a *bhikṣu*, as it was a common habit in the past (although there are still cases of male tonsure masters for women).[17]

Nowadays, the theoretical career for a woman wishing to become a *bhikṣuṇī* in Mainland China thus consists of the following steps:[18]

First, going forth (*chujia* 出家, *pravrajyā*): a female candidate must find a nunnery and have her head tonsured by a tonsure master, most often an *upādhyāyinī*; in many cases, she also receives the ten precepts, even if it is not unusual for *śrāmaṇerikā* ordination to take place at the time of full ordination, in which case the ten precepts are only studied beforehand, in preparation for their formal bestowal.[19]

Second, two-year probationary period (optional): upon reaching age eighteen, the female candidate may receive the six *śikṣamāṇā* precepts, which she must observe for two years; transgressions oblige her to start the probationary period over again;[20] this step is preceded by the *śrāmaṇerikā* ordination.

Third, full ordination (*juzujie* 具足戒, *upasaṃpadā*): from age twenty, the candidate can apply for *bhikṣuṇī* ordination, which involves the dual procedures and, as part of the Triple Platform Ordination, is preceded by *śrāmaṇerikā* ordination (or its renewal, in the event that the candidate has already received it) and followed by the bestowal of the Bodhisattva precepts (which turns the newly ordained into a 'Mahāyāna *bhikṣuṇī*').

In the present article, I will trace the roots of the restoration of Dual Ordinations during the Republican era and provide an account of the early history of these procedures in the PRC. Due to her key role in the process, I will also present Longlian's view about *bhikṣuṇī* ordination. Fur the purpose of this study, I refer only to Mainland China (for Taiwan, see Li Forthcoming), from the 1930s to the beginning of the twenty-first century. The contemporary legacy of these ideas and events in the PRC is discussed in Amandine Péronnet's contribution in this Special Issue. My objective is to probe the historical and ideological context in which the Dual Ordination system was revived, in an attempt to explain how its promotion ended up in strengthening the role of Chinese *bhikṣuṇī*s. As we will see, certain prominent modern Chinese *bhikṣuṇī*s (as well as their male masters) affirmed soteriological gender equality, but also embraced forms of gender asymmetry between female and male Buddhist monastics by reviving the eight *gurudharma*s and the Dual Ordination system, which entails an additional probationary step for female monastics and mandates the presence of both orders at the moment of full ordination for nuns. They endorsed this in the name of orthodoxy (*rufa* 如法, lit. "according to the Dharma") and legitimacy (*hefa* 合法, which more closely refers to a legalistic interpretation of the *Vinaya*) so as to comply with the regulations of the monastic protocol as established by the Buddha. Notwithstanding the implied inequality of an asymmetric treatment of male and female monastics, this ultimately served to raise the status and prestige of *bhikṣuṇī*s in the *saṃgha* and in society.

## 2. The Emergence of the Issue of *Bhikṣuṇī* Ordination in Republican China

*Bhikṣuṇī* ordination and lineages were a debated topic in China during the 1930s and 1940s (Wen 1991, p. 32). Not only did some of the most prominent male *Vinaya* masters of the era interest themselves in ordination procedures, including Dual Ordinations, but the topic was also addressed by scholar *bhikṣuṇī*s trained in the new female Buddhist Academies which offered modern Buddhist education to lay and monastic women in Republican China.[21] The background against which these discussions took place was provided by some modern trends within Chinese Buddhism, i.e., the development of concerns for gender equality (Kang 2016) and the emergence of a *Vinaya* movement, which claimed disciplinary strictness and often took on a text-oriented approach in the name of orthodoxy (Bianchi 2020). In the following section, I will examine both trends, in an

attempt to illuminate how these two separate approaches contributed to the emergence of the Dual Ordination issue.

*2.1. Gender Equality and the Foundation of the* Bhikṣuṇīsaṃgha*s*

The issue of gender equality was raised by well-educated *bhikṣuṇī*s and laywomen.[22] It was part of a larger movement that was questioning the role and place of women in Chinese society at large, and, as Yuan Yuan has demonstrated in her case study on the female Buddhist Academies of Wuhan, it "fitted into the broader women's liberation discourse and the national modernization project" (Yuan 2009, p. 376). These prominent Buddhist women distanced themselves from traditional Buddhist views on females and claimed a leading role for women in the monastic community as well as in society as a whole. To quote Elise DeVido, "they argued that not only do both women and men possess Buddha nature and can become enlightened, but that females should enjoy equality with males whether in the monastic community or in society at large, and women should be liberated from their constraints" (DeVido 2015, p. 78).

Gender equality in Buddhism was also, and indeed first, backed by certain modern male Buddhists. Yang Wenhui 楊文會 (1837–1911), for instance, the well-known layman who initiated some of the most prominent reforms in the field of Buddhist education and publishing, supported *bhikṣuṇī* education, republished many Buddhist scriptures related to women,[23] and advocated a change in the position and role of women within the Buddhist hierarchy (He 1997, pp. 204–5; Valussi 2019, pp. 160–61). Among the scriptures he rediscovered and distributed was the "Biographies of the *bhikṣuṇī*s" (*Biqiuni zhuan* 比丘尼傳, *T* no. 2063), a collection bound to attract much interest within female Buddhist circles as it provided modern *bhikṣuṇī*s with exemplary portraits from the past. Since it also offered details on the history of the foundation and early development of the Chinese *bhikṣuṇīsaṃgha*, this collection became an important reference for the *bhikṣuṇī*s wishing to attest to the legitimacy of their monastic status.[24]

Significantly, the reformist *bhikṣu* Taixu 太虛 (1890–1947), who like Yang promoted the first female Buddhist Academies, was not an advocate of the monastic choice for women, though he maintained that Buddhism did not discriminate against them—as testified by the many enlightened women mentioned in Buddhist scriptures—and encouraged them to serve the Buddhist cause as lay followers.[25] Taixu addressed the topic in a short article published in 1935 in the Buddhist journal *Haichao yin* 海潮音, which deserves to be quoted here as it offers his view about the establishment of the *bhikṣuṇīsaṃgha* in India and about the Buddha's request that *bhikṣuṇī*s respect the eight *gurudharma*s.

Taixu reports some of the complaints he received about the gender inequality embedded among the seven groups of Buddhist disciples (i.e., *bhikṣu*, *bhikṣuṇī*, *śrāmaṇera*, *śrāmaṇerikā*, *śikṣamāṇā*, and male and female lay practitioners, respectively *upāsaka* and *upāsikā*). The argument of Taixu's interlocutor is that ancient Christianity was also unequal towards women, but that in modern times, Christians have developed gender equality, while Chinese Buddhism has not yet adjusted to the equity policies of a modern society. As a result, the interlocutor concludes, in the future there will be no more *bhikṣuṇī*s in China. Taixu replies:

> In Buddhism, there is no inequality between laymen and laywomen, *upāsaka*s and *upāsikā*s . . . But within the monastic community, the gap dividing *śrāmaṇerikā*s, *śikṣamāṇā*s and *bhikṣuṇī*s from *śrāmaṇera*s and *bhikṣu*s is no less than the distance between heaven and the abyss. Why so? The original intention of Śākyamuni Buddha while leading the Buddhist *saṃgha* was that no woman should go forth and join the community, so that the *saṃgha* treasure could be upheld with purity and discipline. But the Buddha's aunt, who had great kindness for the Buddha, strongly insisted on going forth. The Buddha resisted steadily but could not stop her requests and finally imposed strict limitations through the eight *gurudharma*s, also adding some 'secret' precepts. Fundamentally, he wanted to make sure that women knew of the difficulties [of going forth] and encourage [their] withdrawal,

so that his aunt may be the only case capable of becoming a female monastic. This is the reason why, as of today, in Tibet and many other places there are no Buddhist female monastics. ... As for the *bhikṣuṇī* institution, it is absolutely necessary to be strict, first because what was established by the Buddha cannot be changed, and second because if no woman ever became a *bhikṣuṇī* again, this would be fully in compliance with the Buddha's intention.

佛教中在家男女之優蒲塞夷，絕無何不平等處 ... ... 然在出家僧團中之沙彌尼、式叉摩那尼、比丘尼，以視沙彌、比丘，誠不啻天淵之隔。若云何以致此？則釋迦佛原意，住持佛教僧團中，誠不欲有女子出家來加入，以成其純淨律儀之住持僧寶。無如與佛有大恩之姨母强求出家，力拒不絕，乃嚴限制以八敬法且加密戒條，本在令知難而退；或使能出家為尼者，絕無僅有 ... ... 至比丘尼制則斷斷乎須嚴格，一因佛制不可改，二因若能沒有女人作比丘尼，尤合佛心也。
([Taixu 1935](#))

Taixu's understanding of the story of Mahāprajāpatī is in line with a received tradition, which "tells us that soteriologically women are not inferior to men. Socially and institutionally however they are" ([Heirman 2001](#), p. 284). Taixu seems to blame the Buddha's foster mother for having forced the Buddha into creating a female monastic order, which he would rather have avoided. Finally, Taixu proves to be aware of the absence of a *bhikṣuṇīsaṃgha* in other Buddhist traditions, including the Tibetan, and seems to wish the same for China.[26] In this light, the strict respect of the eight *gurudharma*s, including the rules regarding *śikṣamāṇā*s and Dual Ordinations, is given as unavoidable.

A different reading of the *gurudharma*s was provided by Hengbao 恒寶, a prominent scholar *bhikṣuṇī* from Wuhan. Hengbao, the founder and abbess of the Wuhan Pure Bodhi Vihāra (Puti jingshe 菩提精舍), published an article in 1937 on "The Buddhist view on women" (*Fojiao nüxing guan* 佛教女性觀) in the "Dedicated Journal for Female Buddhists" (*Fojiao nüzhong zhuankan* 佛教女衆專刊), the first journal for female Buddhists in China ([Hengbao 1937](#), p. 19), in which she explains that the *gurudharma*s were conceived by the Buddha not because of an alleged discrimination against women, but as a response to the social conditions of the time, "for the sake of [dispelling] oppositions and criticism" ([Yuan 2009](#), p. 389).

This single issue of the "Dedicated Journal for Female Buddhists" (the publication was discontinued because of the Japanese occupation) collected a number of essays by Wuhan *bhikṣuṇī*s, some of which address, more or less directly, female ordination. In her article, Hengbao herself recalls the history of the foundation of the female monastic order by the Buddha, revealing her acquaintance with many canonical versions of the event. Instead of only mentioning Mahāprajāpatī's insistence and the Buddha's final surrender, Hengbao enriches her narrative with many details, casting a nuanced, if not positive light on it. In the received narrative, the role of the Buddha's disciple Ānanda is prominent; Hengbao reports Ānanda's mention of the kindness professed by Mahāprajāpatī towards Śākyamuni Buddha, the statement by the Buddha that women can achieve the four fruits of the path (i.e., stream-entry, once-returning, non-returning, and *arhat*), and the fact that Buddhas in the past had also four assemblies of disciples, i.e., *bhikṣu*, *bhikṣuṇī*, *upāsaka* and *upāsikā*, etc.[27] Hengbao also explains that, according to the *Vinaya* commentary *Shanjian lun* 善見論 (*T* no. 1642),[28] Buddhism will still last one thousand years upon acceptance of the eight *gurudharma*s, instead of only half of its due duration after the creation of the *bhikṣuṇīsaṃgha* ([Hengbao 1937](#), pp. 19–20).[29]

As for the Chinese *bhikṣuṇīsaṃgha*, the principal source of inspiration for modern *bhikṣuṇī*s was the "Biographies of the *bhikṣuṇī*s". Changzhen 常真 (1937), for instance, in response to someone asking her information about the "beginning of the Chinese *bhikṣuṇīsaṃgha*", cites Jingjian 淨檢 (ca. 292–361), whose biography is the first in the collection. This article, which is relevant for our topic because it treats both *śrāmaṇerikā* and *bhikṣuṇī* ordinations,[30] reports, nearly entirely and verbatim, the words by the Central Asian master Zhishan 智山, who was consulted on ordination matters by Jingjian's own

master Fashi 法始. The master explains that he could not bestow her ordination as he did not have the full texts of the *bhikṣuṇī* rules; however, he continues:

> To become a monastic, a female has the ten precepts, which she may receive from the *bhikṣu*s. At the same time, however, she should rely on a [female] monastic instructor to be trained in the precepts.
>
> 尼有十戒，必從比丘授，同時就要以和尚傳戒為依止。([Changzhen 1937](), p. 68)[31]

Jingjian went forth and received the *śrāmaṇerikā* ten precepts together with twenty-four other candidates from Zhishan. Later, a text of rules and procedures for *bhikṣuṇī*s (from the *Vinaya* of the Mahāsāṅghika) reached China and was translated into Chinese. Thus, in the year 357, Jingjian and four of her fellow sisters received full ordination by the foreign *śramaṇa* Tanmojieduo 曇摩羯多 in Luoyang.

In reporting the case of Jingjian, described as the "first Chinese *bhikṣuṇī*", Changzhen shows acceptance of the validity of *śrāmaṇerikā* and *bhikṣuṇī* ordinations that are only bestowed by *bhikṣu*s, but she also points to the need of an *upādhyāyinī* for training a candidate in the precepts. At the same time, she also mentions that only ordinations conferred by the two assemblies should be considered fully legitimate ([Changzhen 1937](), p. 68).

[Hengbao]() ([1937](), p. 20), on the other hand, records the first Dual Ordination, celebrated in the mid fifth century by *bhikṣuṇī*s from Śrī Laṅkā, as the "beginning of the Chinese *bhikṣuṇīsaṃgha*".[32] She compares this first "formal" (*zhengshi* 正式) event with cases that occurred during earlier centuries, when Chinese women—such as Apan 阿潘, according to later sources the first female Buddhist monastic in Chinese history[33]—could only engage in monastic life through the practice of taking the three refuges, and could hence not be called *bhikṣuṇī*s. Implicitly, Hengbao also points to the irregularity of the one-*saṃgha* ordinations, such as Jingjian's, that were taking place before the first Dual Ordination.

The two articles by Hengbao and Changzhen show an interest in the origins and history of the *bhikṣuṇīsaṃgha* in India and China, as seen through a gender-equality concern and in search of exemplary figures from the past to look up to. The above quoted passages also convey a growing awareness of the legitimate procedures for female ordinations, a topic that was being debated by the most prominent *Vinaya* masters of the era.

### 2.2. The Vinaya Movement and Bhikṣuṇī Ordination

The Republican era was also a time of *Vinaya* resurgence ([Bianchi and Campo Forthcoming]()), and many insisted on the establishment of legitimate ordination procedures, i.e., procedures that were believed to have been stipulated and regulated by the Buddha himself in the *Vinaya* literature ([Bianchi 2017b](), p. 116). The irregularity of the ordination system for female monastics when compared with the requirements set up by the *Vinaya*s appeared evident to the *Vinaya* masters of the era, leading to a rediscovery of Dual Ordinations and, even if only to a minor degree, of the figure of the *śikṣamāṇā*.

For instance, in a lecture focused on monastic precepts, *Vinaya* master Hongyi 弘一 (1880–1942) addressed both questions of the legitimacy of the female full ordination processes and of the lack of *śikṣamāṇā*s in Chinese Buddhism. Hongyi notes that the figure of the *śikṣamāṇā* was not known in China during his time, so much so that in the regions north of the Yangtze river, people mistakenly called unmarried Buddhist laywomen by that term. According to *Vinaya* rules, he clarifies, after receiving the ten precepts and at the age of eighteen, a *śrāmaṇerikā* should receive the *dharma* of a *śikṣamāṇā*, which lasts for two years and involves the study of the four *pārājika*s (grave offenses, ultimately corresponding to the first four of the six rules), of the six rules specific to this figure (*liufa* 六法), and of the other *Vinaya* rules and rituals.[34] At the end of the (unbroken) two-year training period, when she turns twenty and reaches the age for full ordination, a *śikṣamāṇā* can receive *bhikṣuṇī* ordination. As for the latter question, Hongyi recognizes that:

> According to the Buddhist system, *bhikṣuṇī* ordination should be taken twice: first, "basic *dharma*" is bestowed by the *bhikṣuṇī saṃgha*; then, the *bhikṣu saṃgha* is

invited to bestow formal ordination. The precepts are properly received only at the time of the formal ordination by the *bhikṣu*s. However, this procedure has no longer been applied since the Southern Song dynasty [1127–1279].

依據佛制，比丘尼戒要重覆受兩次；先依尼僧授本法，後請大僧正授，但正得戒時，是在大僧正授時；此法南宋以後已不能實行了。 (Hongyi 1935)

Remarkably, as Birnbaum (Forthcoming) has pointed out, on the same occasion Hongyi also came to question the legitimacy of the ordination lineage of the Chinese *bhikṣusaṃgha*.[35]

It is clear from the above that Hongyi was well aware that in the case of *bhikṣuṇī* ordination, the procedures prescribed by the *Vinaya* texts had not been implemented for at least one millennium. To my knowledge, however, he did not try to revive them.[36] The two *Vinaya* masters who played a decisive role in the actual implementation of the Dual Ordination system were Cizhou 慈舟 (1877–1957) and Nenghai 能海 (1886–1967).[37] Both masters planned to revive it, but "failed as conditions were not yet ripe at their time" (Zongxing 2019, p. 74). Their legacy was nevertheless taken over after the Maoist era by their two major female disciples, i.e., Tongyuan and Longlian, respectively.

Among the female disciples of *Vinaya* master Cizhou, *bhikṣuṇī*s Kaihui 開慧 and Shengyu 勝雨, together with Yinhe 印和, spearheaded the restoration of Beijing Tongjiao nunnery 通教寺 in 1941. Tongjiao nunnery, once a Ming dynasty temple, soon became a renowned and active *Vinaya* nunnery under the influence of Cizhou (DeVido 2015, p. 81). There they founded the Bajing Xueyuan 八敬學苑, a Buddhist school for *bhikṣuṇī*s significantly named after the eight *gurudharma*s. Tongyuan, who had received tonsure from master Cizhou in 1941 and had taken up residence in Tongjiao nunnery, was trained in this environment and was greatly influenced by both Cizhou and Kaihui, her principal female master, who was particularly engaged in establishing the Dual Ordination system (Wen 1991, p. 32).

According to his disciple Daoyuan 道源 (1900–1988),[38] Cizhou did confer Dual Ordination twice, in 1947 and 1955, at the Anyang Vihāra 安養精舍 in Beijing.[39] Tongyuan was reportedly involved in one of these events (Zongxing 2019, p. 74). To my knowledge, no other source confirms this information. However, considering the above, it is highly probable that Cizhou trained his female disciples from Tongjiao nunnery, including Tongyuan, for Dual Ordinations, irrespective of whether the ordination had taken place or not. Apparently, Tongyuan's knowledge of the procedures was so profound that Longlian decided to involve her as principal master of the discipline in the ordination she organized at the beginning of the 1980s.

Longlian, on the other hand, was introduced to Dual Ordination procedures by Nenghai. In terms of *Vinaya*, this Sino-Tibetan master referred to the Dharmaguptaka tradition (Bianchi 2021a). He insisted that all his disciples follow the rules equally regardless of their gender. However, at the same time he recognized gender asymmetry and stressed gender separation in his communities.[40] In his words:

Male and female *saṃgha*s differ in nature and appearance, they differ in mind and action, thus the precepts must also be different. In reality, should there be no difference, then there is actual inequality, preventing us from seeing the great wisdom of the Buddha.

二部性相不同，心行不同，故戒亦應有別。若無分別，即真不平等，亦不足以見佛之大智慧也。 (Zongxing 2019, pp. 74–75)

In his rigorous approach to the *Vinaya*, Nenghai believed that "women should study as *śikṣamāṇā*s and respect the six rules for two years" (有女須正學六法二年持)[41] and, like Cizhou, urged the re-establishment of Dual Ordinations. In 1937, Nenghai was organizing a *bhikṣuṇī* ordination in the Wutai mountains, but he decided to postpone it because, as stated in his "Notes on ordinations" (*Chuanjie tonggao* 傳戒通告), he realized that "conditions for a proper *bhikṣuṇī* ordination were not yet ripe" (Ma 2015, p. 58). As his disciple Renjie 任傑

reported, a few years later, Nenghai expressed his concern about the fact that *bhikṣu*s were unable to instruct *bhikṣuṇī*s after ordination with these words:

> For a *bhikṣu* to bestow ordination to the *bhikṣuṇī*s and fail to instruct his (female) disciples after doing so is contrary to the Buddha's system and does not protect the Dharma.
>
> 比丘 ... ... 傳比丘尼戒，傳戒後又不能教誡弟子，有違佛制，護法不容！(Ren-jie 1987, p. 70)

We can thus infer that at the basis of Nenghai's interest in the Dual Ordination procedures there was both a concern about orthodoxy (compliance with the *Vinaya* scriptures) and a concern for the proper training and education of the *bhikṣuṇī*s once admitted into the *saṃgha*, which could only be carried out by a female master, or *upādhyāyinī*.[42] This process of reinstating the role of the *upādhyāyinī*, who should guide a female monastic for a period from her tonsure to the two years after ordination, was carried on by Longlian and has become a common practice in the present day PRC.

For the purpose of organizing a Dual Ordination, Nenghai invited the *Vinaya* master Guanyi 貫一 (1875–1954), abbot of the Baoguang monastery 寶光寺, to instruct *śrāmaṇerikā*s on the ordination procedures at Tiexiang nunnery in October 1948. Significantly, Nenghai asked Longlian, who at that time was residing at Tiexiang nunnery and had attended Guanyi's lectures, to impart the *śikṣamāṇā* ordination to the resident monastics; since she was monastically too young (she had only been ordained for eight years, instead of the required twelve), however, in the end Guanyi acted as master of the discipline (Qiu 1997, p. 239).[43] For Nenghai, this had to be the first step on the two-year path to full ordination. Longlian was chosen as the principal master for bestowing the precepts at the upcoming ordination ceremony (Dingzhi 1995, p. 37). It was the eve of the foundation of the PRC, and Nenghai's plan failed as the ordination was ultimately not carried out.[44] But he did not give up and tasked Longlian to take care of "resurrecting" (*huifu* 恢复) Dual Ordinations in the new-era China (Qiu 1997, p. 183). Due to historical circumstances, she was not able to do so before the early 1980s.

To sum up, Tongyuan and Longlian were instructed in Dual Ordination procedures by their own masters, both of whom were *Vinaya* experts, well before the two *bhikṣuṇī*s first met in Beijing's Tongjiao nunnery in 1955. That encounter signaled the beginning of their thirty-six-year-long friendship and created suitable conditions for their cooperation in establishing Dual Ordinations after the Maoist era. Through the establishment of legitimate ordination criteria, as well as the foundation of Institutes of Buddhist Studies, Tongyuan and Longlian significantly contributed to the evolution of the role and status of Buddhist *bhikṣuṇī*s in contemporary Mainland China.[45]

## 3. Assessing the Significance of Dual Ordinations in Post-Mao China

As we have seen, the first Dual Ordination of the modern era in Mainland China was organized by Longlian in Chengdu in the year 1982. This ordination involved only nine *śikṣamāṇā*s; in March 1987, twenty more *śikṣamāṇā*s, graduates from the Institute of Studies directed by Longlian at Tiexiang nunnery, took their turn.[46] As for Tongyuan, she organized the second Dual Ordination in 1984 at the Upper Huayan monastery 上華嚴寺, in Datong (Wen 1991, p. 33). Dual Ordination procedures gradually spread and eventually became the only legal system for *bhikṣuṇī* ordination in the PRC in the year 2000, serving as an integral part of the Triple Platform Ordination system.[47] In this concluding section, I will introduce some of the major aspects of the establishment of Dual Ordinations in the PRC since the 1980s, referring only to the Taiwanese case when it is relevant for the Mainland developments.[48] On the Mainland, the two *bhikṣuṇī*s who succeeded in advocating legitimate female ordination during the Republican era and in implementing them in post-Mao PRC were Tongyuan and Longlian. However, Tongyuan chose to keep a 'low-profile', and we do not have much information about her views. As she refrained from writing about her interpretation of Buddhist doctrines and practices, her opinions about

*bhikṣuṇī* ordination must be inferred from her actions and others' accounts.[49] Longlian, by contrast, was a very influential scholar *bhikṣuṇī* who authored numerous essays and books and frequently gave public talks and interviews; she also played a prominent political role as the first woman to hold a leadership position in the Buddhist Association of China (BAC). For this reason, I will focus mainly on Longlian in the following section.

*3.1. "Resurrecting" Dual Ordinations in Mainland China*

Longlian was officially assigned the task of organizing the first *bhikṣuṇī* ordination of the new era in 1981, after a ban on ordinations that had lasted twenty-five years.[50] The task included the use of Dual Ordination procedures. This assignment came after she had formally requested to "resurrect" Dual Ordinations at the fourth meeting of the newly restored BAC (December 1980). Along with her renown as a scholar *bhikṣuṇī* and her political influence, Longlian's knowledge of the English language may have influenced the decision to involve her in this task, since, as will be explained below, PRC political authorities also perceived this ceremony as an attempt to re-establish the Sinhalese *bhikkhuni-saṅgha*, an example of the so-called 'Dharma diplomacy', i.e., the use of Buddhism for the development of international relations.

At the BAC meeting, Longlian met Tongyuan for the first time after the Cultural Revolution. She told her friend about her intention to resurrect Dual Ordinations, and the latter fully agreed with the plan (Qiu 1997, p. 174). In 1981, Longlian exchanged letters with Tongyuan, seeking her opinion and discussing the contents of and strategies for the ceremony (Zongxing 2019, p. 74). As a consequence, Tongyuan was appointed main *bhikṣuṇī* master of the discipline (*heshang ni* 和尚尼) at the upcoming ordination. The Dual Ordination ceremony took place in January 1982 in Tiexiang nunnery, where the precepts were conferred by ten *bhikṣuṇī* masters, and in Wenshu temple, where they were conferred by the *bhikṣu* masters.

The ordination announcement, published by the official journal of the BAC in February 1982, deserves to be quoted in full:

> Recently, the Wenshu temple in Chengdu, Sichuan Province, held a Dual Ordination ceremony for female candidates. The Buddhist Association of China has expressed its admiration. According to the Buddhist ordination rules, a female candidate must be ordained by the two assemblies before she can become an orthodox (*rufa*) *bhikṣuṇī*. In China, Dual Ordinations were first celebrated in the mid-fifth century in Nanjing by nineteen *bhikṣuṇī*s from the Kingdom of Ceylon (present-day Śrī Laṅkā) [headed by] Tiposalo (the "Biographies of the *bhikṣuṇī*s" name her Tiesaluo). The Wenshu temple Dual Ordination began on 9 December 1981, when the candidates entered the hall to study the rules and rituals. A total of twenty-one monastics participated in the ordination (including nine candidates and twelve advanced *bhikṣuṇī*s). The *bhikṣu* masters included Kuanlin 寬霖 as "master of the discipline", Xinji 心極 as "master of the formal act", Puchao 普超 as "instructor" and Chuanhua 傳華 and others as "witnesses". The *bhikṣuṇī*s had Tongyuan as "master of the discipline", Longlian as "master of the formal act", Dingjing 定靜 as "instructor" and Guojie 果戒 and others as "witnesses". The ceremony lasted for forty days and was successfully completed on 18 January 1982.

> 最近，四川成都文殊院為出家的女眾舉行了一次"二部僧授戒"法會。中國佛協曾致電表示贊嘆。按照佛教授戒法的規定，出家的女眾必須從二部僧授戒後，才能成為如法的比丘尼。我國自公元五世紀中葉師子國 (今斯里蘭卡) 提婆薩羅 (《比丘尼傳》作鐵薩羅) 等十九位比丘尼法師在南京首次實行二部僧授戒。文殊院這次舉行的二部僧授戒法會，從1981年12月9日開始進堂學習律儀。參加受戒的尼眾共21人 (其中新戒9人，增戒12人)。比丘僧由寬霖任得戒師，心極任揭磨師，普超任教授師，傳華等七人任尊證師; 比丘尼僧由通願任得戒師，隆蓮任揭磨師，定靜任教授師，果戒等七人任尊證師。法會歷時40天，至1982年1月18日圓滿結束。(Fayin 1982, p. 21)

The approval by the BAC and the implication that only Dual Ordinations should be considered fully orthodox and legitimate are both remarkable and suggest decades of germination of these ideas.

The textual reference employed was the aforementioned "Dual Ordination Procedures" written at the beginning of the Qing dynasty by the *Vinaya* master Shuyu (*X* no. 1134). As we have seen, they prescribe Dual Ordinations within the Triple Platform Ordination system. For the occasion, Longlian and Tongyuan adapted the text to a modern context. Longlian even had these procedures translated into English for the Sinhalese nuns that were supposed to participate in the ordination.[51] Significantly, the text employed by Longlian and Tongyuan addressed candidates as *śikṣamāṇā*s (*shichamona* 式叉摩那) throughout the rite, which differs from the canonical version, where the term *śrāmaṇerikā*s is used instead (*X* n. 1134: 731c17).

As a matter of fact, the female candidates involved in this first Dual Ordination (and in the other eight ordination ceremonies organized under the supervision of Longlian) were all *śikṣamāṇā*s.[52] To Longlian, the two-year probationary period as a *śikṣamāṇā* was to be understood as an unrenounceable part of the Dual Ordination process. As noted above, this was the view of her master Nenghai, who instructed her to revive Dual Ordinations by a strict observance of all the rules, including the need for the two-year training period. Longlian had also witnessed the bestowal of the *śikṣamāṇā* ordination to the Tiexiang nunnery *śrāmaṇerikā*s back in 1949, under the supervision of Guanyi and Nenghai himself. As Chiu and Heirman (2014) have already observed, Longlian had a decisive influence on the emergence of the *śikṣamāṇā* stage in Mainland China. Even if this stage has not become compulsory, *bhikṣuṇī* ordination can take place no earlier than after a two-year period of time from one's entrance into the Buddhist order according to contemporary official regulations (while for *śrāmaṇera*s only one year is requested, as there are no male *śikṣamāṇā*s), which implicitly allows for the two-year *śikṣamāṇā* training; accordingly, some *bhikṣuṇī* ordination announcements are explicitly geared to both *śrāmaṇerikā*s and *śikṣamāṇā*s.

As for the 'global' context,[53] it should be clarified that the first Dual Ordination of the modern era was held in Taipei in the year 1970. As Yu-chen Li has shown, this system became a widespread ordination criterion for female monastics in Taiwan after 1976 and soon resulted in the Taiwanese *bhikṣuṇī*s' involvement in the international restoration of the *bhikṣuṇī* lineages (Li Forthcoming). At the onset, this seemed to be the case as well in Mainland China, considering that the decision to hold a Dual Ordination ceremony in 1982 was also meant to involve a group of female monastics from Śrī Laṅkā. In an interview, Longlian traced the roots of this plan back to Zhou Enlai 周恩来 (1898–1976), who during an official visit to the South Asian country—he visited Śrī Laṅkā twice, in 1957 and 1964—discovered that female full ordination had disappeared from Śrī Laṅkā and reportedly decided, along with the local authorities, to re-establish the Sinhalese *bhikkhunī* lineage through the intervention of Chinese *bhikṣuṇī*s (Chang 2019, p. 159).[54] Longlian first heard of this possibility in 1980, when she received a visit from a China-based professor of Sinhala (Qiu 1997, p. 239). Professor Lawei 拉維 later published an article in Śrī Laṅkā explaining the history of the introduction of Dual Ordination to China through the intervention of Sinhalese monastics and based on the account provided by Longlian of the relevant passages in the "Biographies of the *bhikṣuṇī*s".[55] Apparently, this article awakened the interest of certain Sinhalese Buddhists. In April 1981, Longlian reportedly met the head of the Ministry of Culture from Śrī Laṅkā in Beijing; on that occasion, it was agreed that candidates from the two countries would be ordained together in Sichuan (Qiu 1997, p. 240). Ultimately, however, the Sinhalese nuns did not attend the Dual Ordination organized at Tiexiang nunnery and Wenshu temple, most likely for political reasons.[56] Nevertheless, the renown of Longlian had already reached far and wide. As a consequence, she was later visited by Karma Tsomo Lekshe, of the Sakyadhita Association of Buddhist Women, who wanted to cooperate with Longlian on an international Dual Ordination ceremony (which also eventually did not happen).[57] As is well known, Sinhalese female monastics, as well as those in other Theravāda countries or in the Tibetan tradition, including Western Buddhist

women, subsequently turned to Korean and Taiwanese *bhikṣuṇī*s, or to different procedures disconnected from the Chinese lineage.[58] However, considering some recent moves by the Karmapa and by masters from the Larung gar in Sertar, the 'global' factor may become relevant again in Mainland China, if allowed (or even favored) at a political level.[59]

### 3.2. Improving the Bhikṣuṇīs' Status: Dual Ordinations from Longlian's Perspective

The establishment of "orthodox" (*rufa*) and "legitimate" (*hefa*) *bhikṣuṇī* ordination procedures in China was an aspiration that Longlian cherished throughout her life. Despite this, to my knowledge, Longlian did not publish any writing on the ordination procedures or their meaning in any of her numerous volumes and essays. However, not only did she often talk about Dual Ordinations with her students and in public, she also gave various interviews touching on this topic, which were recorded by journalists or authors[60] or videorecorded to be included in documentary films.[61] On these occasions, Longlian explained her ideas in terms that strike us as both modern and conservative at the same time. On the one hand, she affirmed gender equality on soteriological grounds (women, she claimed, can become Buddha). On the other, she accepted forms of asymmetry in ordination practices. Nevertheless, in her view this was meant to enforce the legitimacy of Chinese *bhikṣuṇī* ordination from the point of view of orthodoxy, thus indicating that she viewed the introduction of Dual Ordination procedures as another way to improve the status of *bhikṣuṇī*s within the *saṃgha* (Qiu 1997, p. 235).

In a series of interviews she gave to Shanshan Qiu 裘山山, the author of her bestselling biography, Longlian expresses her aspiration to re-establish a legitimate *bhikṣuṇīsaṃgha* and her views on gender equality and asymmetry (Qiu 1997, pp. 284–86).[62] Qiu records an excerpt of an interview, in which she openly asked Longlian about gender equality, stating that in her view there is still an idea of male superiority (*nanzun nübei* 男尊女卑) in Buddhism. Longlian agrees that there is asymmetry in the *Vinaya*, noting as examples ordinations (which must involve both *saṃgha*s for female candidates) and the number of precepts (of which there are ninety-one more precepts in the *Prātimokṣa* for the *bhikṣuṇī*s than in that for the *bhikṣu*s). She then explains that Buddhism teaches the equality of all beings. However, Longlian states, it is not possible to be absolutely equal in this world, where for every person who walks in front or sits in a more elevated seat, there has to be someone else walking behind or sitting lower. In this way, Longlian affirms gender equality in soteriological terms, but allows for the existence and necessity of gender asymmetry in the world we live:

> There is indeed a division between men and women in Buddhism, so that there can be stability within the monastic community. . . . Male supremacy in the secular world is bound to be reflected in the religion.
>
> 佛教中的男女確有一個高下之分，這樣僧團內部才能穩定 . . . . . . 俗世間的男尊女卑，必然會反映到宗教中來。

In the same interview Longlian also explains the position of *bhikṣuṇī*s in the *saṃgha* by recalling the narrative of the foundation of the *bhikṣuṇīsaṃgha*, an argument which is reminiscent of the ideas expressed by the Wuhan nuns of the Republican era, including the belief that observance of the *gurudharma*s will prevent Buddhism from disappearing from the world:

> In the beginning, Śākyamuni Buddha was reluctant to allow women to go forth. . . . The founder of our *bhikṣuṇī* order, the Buddha's aunt, Mahāprajāpatī, was very determined to go forth. The Buddha said, if you insist on going forth, you must observe the eight *gurudharma*s, namely, to have respect for *bhikṣu*s and to observe eight special precepts. In this way, the Dharma will not be destroyed in the future. In order to go forth, Mahāprajāpatī agreed without hesitation. The Buddha built a temple specifically for Mahāprajāpatī (Daaidao), and this is the origin of the name of our Aidao hall. Since *bhikṣu*s came first and *bhikṣuṇī*s later, some phenomena can be easily explained.

當初釋迦牟尼佛是不願意讓女人出家的 ... ... 我們比丘尼的始祖，也就是佛的姨媽大愛道，當初堅決要求出家。佛就説，如果你一定要出家，就必須遵守"八敬法"，即對比丘懷有敬意，遵守八項特殊的戒律。這樣將來才不致毀滅佛法。大愛道為了出家，毫不猶豫地答應了。佛陀就專門為大愛道修建了一座廟，這也就是我們愛道堂名稱的來歷。既然是先有比丘後有比丘尼，有些現象也就好解釋了。(Qiu 1997, p. 285)

Longlian also addressed the issue of Dual Ordinations during an interview recorded on the occasion of an ordination ceremony in 1994 in Aidao nunnery and included in two documentary films on her life (Aidaotang 2002, 2009). The documentary films also include videos showing images that reference the 1982 ordination held in Wenshu temple and Tiexiang nunnery. The interview is translated in the Appendix 70. To sum up, in the interview Longlian clarifies the following points:

- The presence of a *bhikṣuṇīsaṃgha* is important to meet the standards of an ideal Buddhist country.
- A female Buddhist wishing to go forth needs to be instructed by an *upādhyāyinī*, who should follow her from tonsure to the period after full ordination.
- Acceptance by an *upādhyāyinī* and by a certain *bhikṣuṇī* community is a fundamental requirement for full ordination to take place, as stated in *Vinaya* texts.[63]
- Dual Ordination procedures are the result of a gradual process. In the beginning, *bhikṣuṇī*s were ordained following the same procedures as male candidates. Later it was decided that female candidates and newly ordained *bhikṣuṇī*s needed to be instructed by other *bhikṣuṇī*s; but since *bhikṣuṇī*s were not acquainted with the outside world, it was deemed necessary to also involve the *bhikṣu* community.
- Formally a female candidate is ordained only after "ascending the ordination platform" (*dengtan* 登壇) of the *bhikṣu*s (a point which was also made by Hongyi), but the preparatory step at the *bhikṣuṇī* platform is equally necessary.
- Ordinations conferred by only one of the two assemblies, though they were historically considered valid, are not fully legitimate.

In line with the views of Nenghai, Longlian believed that the involvement of the *upādhyāyinī* and the other *bhikṣuṇī*s in the various steps of the ordination process was meant to allow female Buddhists to be duly instructed before and after ordination within a system which emphasizes gender separation. As for the going forth rituals, the first documentary film (Aidaotang 2002) features Longlian performing the *pravrajyā* ceremony (from tonsure and wearing of the *kaṣāya* or monastic robe to the bestowal of the *śrāmaṇerikā* precepts), where she acts as *upādhyāyinī*, thus re-establishing the habit that this step should involve a female master (a difference from her own *pravrajyā*). In Dual Ordinations, the reinstatement of the step involving the *bhikṣuṇī*s is deemed necessary in terms of post-ordination training for the newly ordained, while the role of the *bhikṣu* masters in the process, rather than implying an agenda to exert control over the female order, is instead presented as both a consequence of historical circumstances and the result of the Buddha's concern for the *bhikṣuṇī*s' safety, as explained in some passages of the *Vinaya* texts.[64]

Significantly, the later documentary film removes the emphasis placed by Longlian on the non-legitimacy of *bhikṣuṇī* ordinations conducted by *bhikṣu*s (or *bhikṣuṇī*s) alone. In Chinese Buddhism, be it in Mainland China or Taiwan, this was and still is a very sensitive topic. Taken literally, Longlian's statement that "ordinations conferred at only one place should not be considered legitimate" would imply the fundamental illegitimacy of the Chinese *bhikṣuṇī* lineage as a whole. In reality, following the Chinese *Vinaya* tradition, Longlian believed that ordinations conducted by *bhikṣu*s alone could be accepted. As we have seen, this involved only a minor offense by the *bhikṣu*s bestowing ordination, without invalidating the ordination of the *bhikṣuṇī*s. Accordingly, Longlian admits that an ordination held by only the ten *bhikṣu*s "generally counts as ordination".

Comparisons with modern political concepts ("democracy") and to party administration ("preparatory party member") included in the interview (see full translation in

the Appendix 70) reveal the influence of the PRC's ideological atmosphere on Longlian. Her concerns for gender issues as well as for procedures to be traced back to Śākyamuni Buddha, on the other hand, allow us to connect her with the spread of modernist ideas during the Republic of China, ideas which include an emphasis on gender equality and an attempt to retrace 'original' teachings of the Buddha.[65]

### 4. Conclusions

Both the Dual Ordination system and the related figure of the *śikṣamāṇā* are included in the eight *gurudharma*s, which among other rules also state that a *bhikṣuṇī* must pay obeisance to a *bhikṣu* regardless of his age, or that a *bhikṣuṇī* may not admonish a *bhikṣu*, whereas a *bhikṣu* may always do so.[66] Does the (re-)establishment of these procedures in modern China imply a reiteration of the very idea of *bhikṣuṇī*s' subordination to the *bhikṣusaṃgha*?

In my opinion Longlian, the main character in this story—an exceptional *bhikṣuṇī* who managed to cope with modernity while complying with tradition, and who has become a true symbol of gender empowerment within the *saṃgha*[67]—was not attempting to promote gender inequality through the establishment of the (asymmetric) Dual Ordination system and the figure of the *śikṣamāṇā*. Longlian's main concern was to reinstate the dual procedures in order to make the whole ordination system more legitimate and orthodox, which ultimately also resulted in the improvement of the status of *bhikṣuṇī*s within the Buddhist *saṃgha* and society as a whole. Interestingly Longlian, while consistently rejecting views of gender inequality from a soteriological perspective, took from *Vinaya* master Nenghai the idea of the need for gender asymmetry, which is explained as a consequence of historical and social factors. In this light, male masters are involved in the ordination process in order to protect (rather than to control) the *bhikṣuṇī*s, an argument that was made also in the *Vinaya* of the Dharmaguptaka.[68]

In a nutshell, I believe that the establishment (or "resurrection", as it is usually termed) of Dual Ordinations in modern China should be seen as the result of a few seemingly unrelated phenomena. First of all, from its onset during the Republican period, the idea of establishing Dual Ordinations was connected with a modern notion of orthodoxy, which was notably searched for in the scriptures, and involved the adoption of a text-oriented approach to Buddhist practices. In the eyes of many modern Chinese *bhikṣuṇī*s, this idea was also related to the search for legitimacy of their monastic status, in the wake of modern perspectives on gender equality within the Buddhist community. In later years, a third aspect emerged, as the Dual Ordination system assumed a 'global' dimension and was connected with the re-establishment of the *bhikṣuṇī* order within other Buddhist traditions: phenomena that are integral to a modern interpretation of Buddhism.

To conclude, the asymmetry embedded in the ordination system was endorsed in modern times in the name of legitimacy/orthodoxy, seemingly without advocating ideas of inequality within the *saṃgha*. On the contrary, considering that both *śrāmaṇerikā* and *bhikṣuṇī* ordinations were bestowed by male masters for centuries within Chinese Buddhist monasticism, the involvement of female masters resulted in a form of female empowerment, if not in full-fledged equality.

**Funding:** This research was funded by the Department of Philosophy, Social Sciences and Education, University of Perugia: Ricerca di base (2018).

**Acknowledgments:** I wish to thank Nicola Schneider for first inviting me to participate in the workshop on "L'asymétrie sexuelle dans les différentes traditions bouddhiques à travers le prisme de l'ordination et de l'éducation des nonnes" (Paris, 16 January 2015), where we began our exchanges on the topic of this Special Issue. It was due to her persistence, the consonance of our interests and the stimulating exchange over the years, that we finally proposed this project to the Chiang Ching-Kuo Foundation of International Scholarly Exchange (CCKF), which gracefully chose to support it with a Conference Grant 2020/2021. My gratitude also goes to all the participants of the conference, "Gender Asymmetry in the Different Buddhist Traditions Through the Prism of Nuns' Ordination and

Education" (Perugia, 16–17 May 2022), for their lively and helpful discussions; I am especially grateful to *bhikkhunī* Dhammadinnā, Daniela Campo, Ann Heirman, Amandine Péronnet, Nicola Schneider, and Alexander Von Rospatt, for reading previous versions of this article and for their comments and feedback. I am also indebted to *bhikṣuṇī* Shi Guoping 釋果平, *bhikṣuṇī* Shih Heng-Ching 釋恆清, *bhikṣuṇī* Shi Hongzhi 釋弘智 and *bhikṣu* Shi Xianshi 釋賢世 for providing insightful suggestions, useful information and precious material.

**Conflicts of Interest:** The author declares no conflict of interest.

## Abbreviations

T   *Taishō* 大正 (*Taishō shinshū daizōkyō* 大正新修大藏經. Edited by Takakusu Junjirō 高楠順次郎 and Watanabe Kaikyoku 渡辺海旭. Tōkyō: 1924–1935).

X   *Xuzangjing* 續藏經 (*Dainippon zoku zōkyō* 大日本續藏經. Edited by Maeda Eun 前田慧雲, and Nakano Tatsue 中野達. Kyōto: 1905–1912).[69]

## Appendix A. Longlian Explaining Dual Ordinations in 1994[70]

Buddhist disciples are called the "disciples of the four assemblies", and are male and female monastics who went forth and male and female lay householders.[71] Ordained male monastics are called *bhikṣu*s and ordained female monastics are called *bhikṣuṇī*s; those who have not yet received complete ordination but have gone forth are *śrāmaṇera*s and, the female ones, *śrāmaṇerikā*s. Thus, in Buddhism "four groups" means *bhikṣu*s, *bhikṣuṇī*s, laymen and laywomen. Only a place where all four assemblies of disciples are complete is called a "Middle kingdom" (*Zhongguo* 中國). In Buddhism, the special name "middle kingdom" refers to a place that is the center of Buddhism. So, in order to meet this standard, the presence of all four assemblies of disciples is necessary.[72]

佛的弟子，稱為四眾弟子。四眾弟子就是出家男女二眾，在家男女二眾。出家的男眾被稱為比丘，女眾就稱為比丘尼。初出家還沒受大戒的，男的稱為沙彌，女的稱為沙彌尼。佛教當中說四眾弟子就是說的，比丘，比丘尼和在家的男居士，女居士。要四眾弟子齊全的地方，才稱為中國。佛教裡面的特殊名字叫中國，它的意思就是說，這個地方是佛教的中心，那麼要夠得上這個標準，就是要四眾弟子齊全。

Ordination procedures have been established gradually. In order to be ordained, *bhikṣu*s have to undergo a "three-times formal act" (*san fan jiemo* 三番羯磨),[73] and *bhikṣuṇī*s also have to follow the same procedures. It is a democratic procedure; i.e., whenever there is a person wishing to become a *bhikṣu* or a *bhikṣuṇī*, it is necessary to select from the monastic community a group of ten high-ranking *bhikṣu*s of great virtue and appropriate monastic age. They are convened in order to hold this particular examination, which is also called the "ritual of ascending the ordination platform" (*dengtan jiemo* 登壇羯磨). A special place shall be provided,[74] since ordination is a high-level and secret assembly that cannot be attended by everybody. On the ordination platform there are ten persons, the "master of the discipline" (*jie heshang* 戒和尚), the "master of the formal act" (*jiemo shi* 羯磨師), the "instructor" (*jiaoshou shi* 教授師), and seven "venerable witnesses" (*zun zheng shi* 尊證師)—"venerable" because they are high-ranking *bhikṣu*s, while "witnesses" expresses their function as "attestors". The union of these ten persons makes it a high-level assembly, a special assembly. Candidates must receive the approval of this assembly to become *bhikṣu*s or *bhikṣuṇī*s. This assembly cannot be attended by anybody else; the attendants of the principal master and all the other attendants (*yinli shi* 引禮師) are not allowed to take part in it. On the platform, there are only the ten masters who hold this important assembly; in addition, the ordination candidates are also there. This assembly is organized in such a strict and secret way.[75]

傳戒的手續是逐步建立的。那麼比丘也要經過三番羯磨傳戒的手續。比丘尼同樣
要經過手續。這個手續是一個民主的手續，就是說誰要當比丘或者是比丘尼，都
要在僧團當中選出十位地位特別高的，道高德重，戒臘須彌的人，來開這個特
別的審查會，這個就是現在所謂的比丘登壇羯磨。這個開會還要有個特殊的地
方，是一個高級的秘密會議，不得是全體人都來參加。這個壇上就有十位，包括
戒和尚，羯磨師，教授師和七位尊證師。尊就是他有地位，證就是他來證明。這
十個人組合起來，它就成了一個高級的會議，特殊的會議。那麼這個新戒要通過
允許他成為比丘，要經過這個會議。這個會就是說其他人不能參加。戒和尚帶的
侍者和那些引禮師都不能參加這個會。壇上只有這個十師開這個高級會議。另外
就是受戒的新戒在裡面。這個會議的組織就是這樣一個比較嚴密而秘密的一種
會。

As for *bhikṣuṇī*s, in the very beginning they also followed the same ordination procedures (as male candidates). Later, however, it was said that this was not sufficient, because this way a *bhikṣuṇī* only had male masters: how would a male master ever take care of her? For this reason, it was deemed necessary that *bhikṣuṇī*s be instructed and guided by other *bhikṣuṇī*s. Therefore, whenever a woman wishes to enter the monastic order, she must find another woman who will act as her "master of the discipline" (*upādhyāyinī*). The latter will be responsible for instructing the candidate, so as to establish with her a master-disciple relationship (*shitu guanxi* 師徒關係).[76] But after this rule was established, some new problems arose. I.e., since *bhikṣuṇī*s lived in deep seclusion, rarely came out and thus were not acquainted with the outside world, it happened that throughout history some problems arose in the acceptance of new candidates. Therefore, it was understood that it was not sufficient that one be approved only by ten *bhikṣuṇī*s; instead, it was necessary to also be approved by ten *bhikṣu*s, which added a further step.[77]

那麼比丘尼最初也就是這樣受戒的。後來說不行，這個比丘尼的師父都是男
的，哪個去管她呢?所以比丘尼一定要由比丘尼來教導，那麼她要出家的時
候，就要找個女的給她當戒和尚，這個戒和尚就要負責教這個新戒，要建立
起師徒關係。這個規矩建立之後呢?後來又有問題，就是說，在比丘尼，她都
深居簡出，外面多少情況她不熟悉，有時收來的新戒，在這個歷史上就有些問
題。所以說光是十個比丘尼通過還不行，還要十個比丘來通過，這就更進一步。

This led to a dual procedure. A *śrāmaṇerikā-śikṣamāṇā* wishing to receive female full ordination, must undergo a first examination on the *bhikṣuṇī* ordination platform; this way she becomes a "fundamental" *bhikṣuṇī*, called a "basic *dharma bhikṣuṇī*" (*benfa ni* 本法尼).[78] This is like a "preparatory *bhikṣuṇī*", in the same sense as the political title of "preparatory party member" (*yubei dangyuan* 預備黨員). But this phase is very short: it is requested that, on that same day, as soon as the basic *dharma* ordination has been conferred among the *bhikṣuṇī*s, female candidates reach the ten *bhikṣu*s' platform in order to receive the precepts for a second time. This is why it is called "ordination by the two assemblies".

所以這就成了兩道手續，一個女的沙彌正學女，要受比丘尼戒，要經過比丘尼的
壇上十師開會審查了，才是一個基礎的比丘尼，叫本法尼，像是一個預備比丘尼
一樣，就像預備黨員那個意思，但是這個時間很短。它有要求你當天，今天在比
丘尼當中，把這個本法尼戒受了，馬上就在這一天之內，要到這個是個比丘壇上
十師當中去，重受二道戒。所以就稱為二部僧戒。

Ordinations conferred at only one place should not be considered legitimate. But what if the ceremony was held only by the ten *bhikṣu*s? Is that candidate considered to have been ordained or not? Ordinarily speaking, it should count as ordination. Yet, that *bhikṣuṇī* misses the first step of the procedure, her ordination has not been conducted according to the system established by the Buddha, since the part of the procedure involving the approval by the *bhikṣuṇī* assembly is lacking. This is already illegitimate in itself. The second problem is that

this *bhikṣuṇī* did not find a *bhikṣuṇī* master by whom to be instructed into the precepts. This *bhikṣuṇī* was only ordained by the *bhikṣu*s. Buddhism particularly emphasizes gender differences. Hence, even if she has been ordained, this *bhikṣuṇī* cannot follow a male master of the discipline. Therefore, she needs to have an *upādhyāyinī*; only in this way would she be duly instructed. This newly ordained *bhikṣuṇī*, immediately after ordination, needs to follow that female master, and study with her the three Buddhist teachings [i.e., monastic discipline, meditation and wisdom]. For a male master of the discipline, no matter how knowledgeable and virtuous he may be, it would not be easy to provide that mentorship. Therefore, this is how the system was set up. That is, it is not legitimate to bestow ordination without *bhikṣuṇī* masters.[79] Śākyamuni Buddha said that you must first find an *upādhyāyinī* to admit and instruct you, and that you can only be ordained after the *bhikṣuṇīsaṃgha* has acknowledged you and accepted you to live there.

只有一個地方受都是不合法的。但是呢，只有比丘授呢，這個人算不算得戒呢，照理說應該算得戒，但是她就缺了這第一道手續。沒有依照佛的制度，沒有通過比丘尼的會議，這就是第一個不合法。第二個呢，就是說沒有找到比丘尼給她當師父，給她當戒和尚，她光是在比丘當中受了戒下來。佛教特別是男女有別。那麼她就是受了戒下來，也不能跟到這個男的戒和尚，所以她就必須要有一個女的戒和尚，才算是真正教授她的師父。受了戒之後照說這個新戒，就應該跟到這個女的戒和尚，學這個佛法當中的三學。男的戒和尚，道高德重也不好辦，不能管教。所以這個制度是這樣建立起來的，就是說沒有比丘尼的和尚，而授戒呢，不合法。釋迦佛說的，要先把你的比丘尼和尚找了，承認教你。比丘尼的僧團，承認接納你在那裡住，你才能受戒。

Therefore, ascending the ordination platform in the midst of the *bhikṣuṇī*s is a 'preparatory' step, but a necessary one. Only after ascending the ordination platform in the midst of the *bhikṣu*s is it decided that you have eventually become a *bhikṣuṇī*. However, the master of the discipline will also stress that, as you have been ordained there, after ordination you will have to continue studying the precepts with that *upādhyāyinī*. So, this is how this system was established. It is called Dual Ordination.

所以，比丘尼當中登壇算是一個預備，但是是必要的預備。而比丘當中登壇才算是最後，決定你最後成為比丘尼了。但是戒和尚還是說，你在這裡受了戒，以後還是要去跟著你那個比丘尼的戒和尚學習，所以這個制度是這樣建立起來的，稱為二部僧戒。

## Notes

1    A first draft of this article was presented at the conference "Gender Asymmetry in the Different Buddhist Traditions Through the Prism of Nuns' Ordination and Education", co-organized by Ester Bianchi and Nicola Schneider (Perugia, 16–17 May 2022). The two articles by Ester Bianchi and Amandine Péronnet in this Special Issue were originally presented together in an attempt to assess gender asymmetry in Chinese monastic Buddhism in modern and contemporary China, with reference to the issue of ordination. In the present paper, Buddhist terminology is given in Sanskrit.

2    According to the biography of Sengguo 僧果, as recorded in the "Biographies of the *bhikṣuṇī*s" (T no. 2063: 939c–940a), a mercantile ship arrived in China in 429 with a group of *bhikkhunī*s from Śrī Laṅkā on board. Another group of *bhikkhunī*s reached China later, in 433, creating the quorum necessary for full ordination. More than three-hundred Chinese women were thus ordained (or, in many cases, re-ordained) by the Sinhalese *bhikkhunī*s. Although *bhikṣuṇī* ordinations had already occurred in China before, this was the first Dual Ordination. The re-ordination of more than three hundred *bhikṣuṇī*s through this Dual Ordination ceremony is also mentioned in the biography of Huiguo (ca. 364–433) (T no. 2063: 937b18–c7). See Heirman (2001, pp. 275–304), and Zheng (2010).

3    As we will see, *the gurudharma*s were discussed in Buddhist circles during the Republican era. They were also reflected in the name of one of the female Buddhist Academies of Wuhan (Bajing xueshe 八敬學社, mentioned in Yuan 2009, p. 385) and later in Beijing (Tongjiao nunnery 通教寺's Bajing Xueyuan 八敬學苑, which will be addressed below).

4    The eight rules differ partially in the various *Vinaya*s; in the *Vinaya* of the Dharmaguptaka, which was adopted by Chinese Buddhists, they are: (1) Even when a *bhikṣuṇī* has been ordained for one hundred years, she must rise up from her seat when

seeing a newly ordained *bhikṣu*, and she must pay obeisance; (2) A *bhikṣuṇī* may not revile a *bhikṣu* saying that he has done something wrong; (3) A *bhikṣuṇī* may not admonish a *bhikṣu*, whereas a *bhikṣu* may admonish a *bhikṣuṇī*; (4) After a woman has been trained as a *śikṣamāṇā* for two years, the ordination ceremony must be carried out in both orders; (5) When a *bhikṣuṇī* has committed a *saṃghāvaśeṣa* offense (an offense that leads to a temporary exclusion), she has to undergo the penance in both orders; (6) Every fortnight, *bhikṣuṇī*s have to ask *bhikṣu*s for instruction; (7) *Bhikṣuṇī*s cannot spend the summer retreat (rainy season) in a place where there are no *bhikṣu*s; (8) At the end of the summer retreat, *bhikṣuṇī*s also have to carry out the *pravāraṇā* ceremony in the *bhikṣu* order. On the history of the beginning of the *bhikṣuṇī* order, see Anālayo (2016, 2019) and, for a different view, von Hinüber (2008); on the narrative of the foundation according to the *Vinaya* of the Dharmaguptaka and a comparison with the other available *Vinaya*s, see Heirman (2001, pp. 278–84).

5    On *bhikṣuṇī* ordination procedures according to the *Vinaya* of the Dharmaguptaka, see particularly Heirman (2002, vol. 2), and Li (2008). For a critical analysis of historical sources, see Huimin (1999, 2007), Heng-Ching (2000), and Chang (2019). For an overall presentation, see Anālayo (2018).

6    In the *Vinaya* of the Dharmaguptaka, the ten precepts of a *śrāmaṇerikā* are the same as those of a *śrāmaṇera* and read as follows: (1) not to kill; (2) not to steal; (3) not to have sexual intercourse; (4) not to lie; (5) not to take intoxicating substances; (6) not to take part in singing, dancing and other amusements; (7) not to use garlands or perfumes; (8) not to sleep on high or broad beds; (9) not to handle silver or gold; (10) not to eat food out of regulated hours. In China, *śrāmaṇerikā* ordination was often bestowed by a *bhikṣu*, whereas according to *Vinaya* rules an *upādhyāyinī* (female master of the discipline) should be involved (Heirman 1997, pp. 43–44). On *śrāmaṇerikā* ordination, see also Heng-Ching (2000, pp. 509–10).

7    The *Vinaya* of the Dharmaguptaka has 250 precepts for *bhikṣu*s and 348 for *bhikṣuṇī*s. For the *Prātimokṣa*, the set of rules for the *bhikṣuṇī*s, refer to Heirman (2002, vol. 2).

8    The Triple Platform Ordination was first conceived by Guxin Ruxin 古心如馨 (1541–1615) and later elaborated by his first- and second-generation disciples Hanyue Fazang 漢月法藏 (1573–1635) and Duti Jianyue 讀體見月 (1601–1679). This was a time of Buddhist resurgence, and ordination reform was conceived in response to a previous ban on Buddhist ordinations (Lepneva 2022, and Wu Forthcoming). On the *Vinaya* movement of the end of the Ming (1368–1644) and beginning of the Qing dynasty (1644–1911), see also Liu (2008), Sheng-Yen (1991), and Wu (2008).

9    The Southern Monastery (Nanshan 南山) *Vinaya* lineage is traditionally believed to have been founded by Daoxuan 道宣 (596–667) and also includes the *Vinaya* masters who elaborated the Triple Platform Ordination and Dual Ordinations in the late imperial period.

10   Dual Ordinations are still mentioned in historical records of the Tang dynasty (618–907). Note that, as reported by Zanning (*T* no. 2126: 238b24-c8), in the year 972, the Song Emperor Taizu 太祖 (r. 960–975) issued a decree prohibiting *bhikṣuṇī*s from going to male monasteries for ordination, implicitly establishing that ordinations could only be bestowed by *bhikṣuṇī* masters. In the "Complete Chronicle of the Buddha and Patriarchs" (*Fozu tong ji* 佛祖統計), however, Zhipan 志磐 (1220–1275) informs us that this prohibition only lasted a few years (*T* no. 2126: 396b4-9). My gratitude to ven. Xianshi 賢世 for pointing me to these canonical texts. On this issue, also see Heng-Ching (2000), and Huimin (2007).

11   Shuyu was a direct disciple of *Vinaya* master Jianyue (see above, note 8), and his book was written on the basis of a *bhikṣuṇī* ordination performed by Jianyue himself in 1667. In the seventeenth century, *Vinaya* master Hongzan 弘贊 (1611–1685), in his *Biqiuni shoujie lu* 比丘尼受戒錄 (*X* no. 1132) also mentions a Dual Ordination that he organized at the request of numerous female candidates from different places (quoted by Heng-Ching 2000, p. 532).

12   Guṇavarman considered one-*saṃgha* ordination to be legitimate in the absence of the proper conditions for Dual Ordinations. In Huiguo's biography he agrees that Chinese women could be ordained in the same way as Mahāprajāpatī, implying that the two situations were comparable because of the absence of *bhikṣuṇī*s; however, he also stated that whenever the *bhikṣuṇīsaṃgha* is established, the Dual Ordination requirements should be respected (*T* no. 2063: 937b27 and 937c2–3, quoted by Heirman 2001, p. 289). Additionally, in the biography of Sengguo, the ordination of Mahāprajāpatī and the five hundred Śākya women is also presented as a precedent for the first Chinese female ordination (*T* no. 2063: 939c14–21, quoted by Heirman 2001, p. 290). Upon the arrival of the Sinhalese *bhikkhunī*s, Guṇavarman approved the Dual Ordination to take place as a way to augment the value of the first ordination and thus without neglecting its legitimacy. Elsewhere Guṇavarman also advanced the idea that ordinations conferred only by *bhikṣu*s produced a (minor) offence on the part of the *bhikṣu*s without impacting on the *bhikṣuṇī* candidates; later, this point was also made by Daoxuan, who settled the issue for the succeeding centuries. Guṇavarman's opinion is recorded in the *Gaoseng zhuan* 高僧傳 (*T* no. 2059: 34la28–b7, quoted in Heirman 2011, p. 621 n. 62). Daoxuan reiterates this idea in his commentaries on the *Vinaya* of the Dharmaguptaka (*T* no. 1804: 519c–15, quoted in Heirman 2011, p. 621; and *X* no. 728, quoted in Huimin 2007, n. 17). See also Heng-Ching (2000, pp. 522–24).

13   The *śikṣamāṇā* (*shichamona* 式叉摩那, lit. "training oneself", or *zhengxue nü* 正學女, *xuefa nü* 學法女) is a *śrāmaṇerikā* who accepts six precepts for a probationary period of two years. These six precepts map partially onto the ten *śrāmaṇerikā* precepts (see above, note 6) and are: (1) not to have sex; (2) not to steal; (3) not to kill; (4) not to lie; (5) not to consume alcohol; (6) not to eat at improper times. According to Ann Heirman the difference between a *śikṣamāṇā* and a *śrāmaṇerikā* may only be formal, as the admission ceremony for the former is very elaborated, while for the latter no formal act is involved (Heirman 1997, p. 50). Conceived as an evaluation period of the candidate's suitability for full ordination, according to the *Vinaya* of the Sarvāstivāda, the probationary

period was also created to check the possibility of pregnancy in female candidates (Huimin 2007, p. 16; Heirman 2008, p. 108). On *śikṣamāṇā* ordination, see also Heng-Ching (2000, pp. 510–13), and Heirman (1997, pp. 36, n. 14, 45–47).

14   On Longlian, see (Bianchi 2001), Bianchi (2017a), Qiu (1997) and, for a collection of her writings, Wang (2011). On Tongyuan, see Péronnet (2020, pp. 133–35), and Wen (1991). On Longlian and Tongyuan's conjunct roles in the establishment of Dual Ordinations, see also DeVido (2015), and Zongxing (2019).

15   She is termed *xuefa nü* (another name of a *śikṣamāṇā*) or *jinfa nü* 近法女, "female studying/approaching the *dharma*" (Li 2020, p. 601).

16   According to the *Vinaya*, this is a *bhikṣuṇī* who guides and instructs a new candidate from the moment she asks to go forth until two years after ordination (Heirman 1997, p. 44, n. 67).

17   See above, note 6.

18   The present paper is focused on *bhikṣuṇī*s, i.e., fully ordained Buddhist monastics; therefore, I will not discuss the figure of the *caigu* 菜姑 ("vegetarian woman"), i.e., lay Buddhist nuns, or other forms of female Buddhist engagement. On the "vegetarian women", see Ashiwa and Wank (2019); on this and other forms of female Buddhist commitment during the Republican Era, see Li (2020, pp. 591–98). For a woman's monastic career according to the *Vinaya* texts, see Heirman (1997, 34 ff), and Heng-Ching (2000). I am grateful to ven. Guoping 果平 and ven. Hongzhi 弘智 for providing some information and details (WeChat communications, July 2022).

19   The *Vinaya* of the Dharmaguptaka fixes the minimum age for a *śrāmaṇerikā* at twelve.

20   The need to retake the six rules and begin the two-year training all over again in case of violation is a requirement of the *Vinaya* of the Dharmaguptaka (Heng-Ching 2000, p. 512). Since the first four rules correspond to the first four *pārājika*s (offenses entailing irreversible loss of monastic status), their transgression leads to permanent exclusion from the *saṃgha*; the extension of the two-year training is meant as a consequence of the transgression of the fifth and sixth rules (eating at the wrong time or drinking alcohol), and to "offences closely linked to the four *pārājika*s" (Heirman 1997, p. 48).

21   The Wuchang Female Buddhist Institute of Studies (Wuchang foxueyuan nüzhong yuan 武昌佛學院女眾院, renamed in 1931 Shijie foxue yuan nüzhong yuan 世界佛學院女眾院) was founded in 1924 by Taixu 太虛 as part of his Wuchang Buddhist Academy. It was later followed by other similar academies, in Wuhan (e.g., Pure Bodhi Vihāra, founded in 1931, and the Hankou Academy for Nuns at Qiyin nunnery 棲隱寺, in 1948) and throughout the rest of the nation (notably the Academy for Female Buddhists in Fenghua, Zhejiang, and many other locations). See DeVido (2015), and Yuan (2009). On laywomen and *bhikṣuṇī*s during the Republic of China, see also He (1997), and Li (2020).

22   Among the most prominent Buddhist laywomen, who are not dealt with in the present study, was Zhang Ruzhao 張汝釗 (1900–1969), also known as Zhang Shenghui 張聖慧, who wrote many articles on topics related to Buddhism and women in major Buddhist journals (Grant 2017; Yuan 2009, pp. 375–412).

23   In general, these scriptures were revised with a new gender sensibility (Valussi 2019, p. 141). The idea was to demonstrate that Buddhism was already 'modern,' including in terms of gender equality, and thus not to be counted as a reason for China's backwardness.

24   The *Biqiuni zhuan* is a collection of biographies of sixty-five Chinese *bhikṣuṇī*s who lived between the years 335 and 516; it is attributed to Baochang 寶唱, who reportedly compiled it in 517 (Liang dynasty). For a translation, see Tsai (1994). On the reprinting of the "Biographies of the *bhikṣuṇī*s" and the publication of its sequel during the Republican era, see Valussi (2019, pp. 160–61).

25   The same opinion was shared by Yinguang 印光 (1862–1940), the famous Pure Land master. See Valussi (2019, pp. 158–60). On Taixu's thoughts on women, see also DeVido (2015, pp. 75–79).

26   On other occasions, Taixu reformulated this concept in a more nuanced way. For instance, in 1930 he stated that "The reason why the Buddha, when he was in the world, [first] did not allow women to go forth, was due to the heavy responsibility attached to that choice and to the difficulties of the Buddhist monastic life; therefore, he did not allow women to do so lightly. Then, when he was approached by women with a sincere and pure mind willing to go forth, the Buddha listened to their request and gave his approval. Thereupon he taught women widely" (當佛在世時，不許女眾出家，其原因以出家之責任繁重，而梵行亦難實修，未便輕許。嗣有出真誠懇切的清淨心來出家者，佛遂聽許，即以廣為化導一切女人, Taixu[1930] 1980). On these passages, see DeVido (2015, pp. 76–77).

27   These details are reported in many of the narrations about the foundation of the *bhikṣuṇīsaṃgha*. See Heirman (2001, pp. 279–82, Table 1 and Table 2).

28   Abbreviated title for *Shanjianlü piposha* 善見律毘婆沙 (*T* no. 1462), a *Vinaya* commentary whose translation is attributed to Saṃghabhadra (488). It was considered a translation of the *Samantapāsādikā* (a commentary attributed to Buddhaghoṣa) throughout the twentieth century. This attribution has since been questioned by scholars (Heirman 2004).

29   Paraphrase of a passage of the *Shanjian lun* (*T* no. 1462: 796c21–23). The different *Vinaya*s offer different interpretations of this prophecy. See Heirman (2001, p. 281, n. 41).



30   Zhu Jingjian 竺淨檢's biography is included in the "Biographies of the *bhikṣuṇī*s" (*T* no. 2063:934c2–935a5, tr. Tsai 1994, 17–21). The lack of *bhikṣuṇīsaṃgha* involvement in her ordination led to discussions in the mid fourth century, which is also reported in Jingjian's biography. See Heirman (2001, p. 275).

31   The original text in the "Biographies of the *bhikṣuṇī*s" reads: 尼有十戒得從大僧受 。但無和上尼無所依止耳 。撿即剃落從和上受十戒 。同其志者二十四人 (*T* no. 2063: 934c13–15).

32   See above, note 2.

33   Apan is mentioned in the *Dasong sengshilüe* 大宋僧史略 (*T* no. 2126: 237c22–25), the "Song dynasty brief history of the *saṃgha*" by Zanning 贊寧 (919–1001). See Heng-Ching (2000, p. 518).

34   For the six rules, see above, note 13. As for the other requirements of a *śikṣamāṇā*, Hongyi refers to the *Vinaya* of the Dharmaguptaka, which mentions that a *śikṣamāṇā* should study all the *bhikṣuṇī* precepts, except for giving food to a *bhikṣuṇī* and receiving food with one's own hands (*T* no. 1428: 924c2–4, quoted by Heirman 1997, p. 48). See also Huimin (2007, n. 24).

35   Birnbaum (Forthcoming) explains that Hongyi questioned the authenticity of his own ordination and, as a consequence, of the whole Chinese *saṃgha*, because of the fracturing of transmission lineages.

36   On the other hand, Hongyi's formulation of the five precepts was likely meant to provide the *caigu*, "vegetarian women", with a Mahāyāna way to be a "five-precept monastic" (Raoul Birnbaum, personal communication, May 2022). On Hongyi and the vegetarian women, see Li (2020, pp. 599–603).

37   On the role of Cizhou and Nenghai in the *Vinaya* movement during the Republic of China, see Bianchi (2017b), and Campo (2017).

38   Daoyuan studied with Cizhou in the 1940s and later became an influential ordination master in Taiwan; he insisted on following text-informed procedures and supported Dual Ordinations. See Li (Forthcoming).

39   See Daoyuan (1982). There is no reference to these two events in the short biography of Cizhou that the same Daoyuan wrote in 1958 (included in Xincheng 2004, pp. 1–6).

40   On Nenghai's views about gender equality/asymmetry, see Wang and Fu (2017).

41   These two lines are taken from Nenghai's "Ode to the liberation precepts of the seven assemblies of disciples" (*qi zhong biejietuojie lüesong* 七眾別解脫戒略頌), included in Nenghai[1936] (Nenghai[1936] 1995, p. 13) and also quoted by Renxiang (1994, p. 34).

42   The above quoted words are reminiscent of the following passage from the *Vinaya*: "If a *bhikṣuṇī* admits many disciples, but does not tell them to study the precepts for two years and does not give them support in two things, then it is a *pācittika* [an offense that needs to be expiated]" (*T* no. 1428: 760a8–b14, translated in Heirman 1997, p. 77). Another reason for the involvement of the *bhikṣuṇīsaṃgha* in the ordination procedure, which nevertheless is not mentioned by Nenghai or Longlian, is the need to ask female candidates intimate questions which it would be embarassed to answer in front of male masters. My gratitude to ven. Shih Heng-Ching 釋恆清 for pointing this out to me (personal communication, August 2022).

43   The *Vinaya* requests a minimum seniority of twelve years for a *bhikṣuṇī* to act as *upādhyāyinī* (Heirman 2002, p. 89). As is noted above (note 6), in China *śrāmaṇerikā* ordination was often bestowed by a *bhikṣu*, whereas according to *Vinaya* rules an *upādhyāyinī* (female master of the discipline) should be involved (Heirman 1997, pp. 43–44). This may explain why, in the 1948 *śikṣamāṇā* ordination at Tiexiang nunnery, the six precepts were ultimately bestowed by Guanyi, a male master.

44   Note that, according to Wang and Fu (2017, pp. 10–11), the ordination did take place.

45   Aside from establishing Dual Ordination procedures in China, a second lifelong cherished goal of Longlian was the creation of a *bhikṣuṇī* college. She had already advanced the formal request in 1980 at the fourth meeting of the BAC. The Sichuan Nuns' Institute for Buddhist Studies (Sichuan nizhong foxueyuan 四川尼眾佛學院) was officially founded in 1983 inside Tiexiang nunnery. In 2007, the Institute was moved out of Tiexiang nunnery: now under the directorship of Ruyi 如意, one of Longlian's closest students, it is presently located in Pengzhou (near Mianzhu). See Bianchi (2001, 2017a). As for the influence of Tongyuan on the establishment of the Mount Wutai Buddhist Institute for Nuns (Zhongguo Wutaishan nizhong foxueyuan 中國五台山尼眾佛學院), see Péronnet (Forthcoming).

46   Longlian organized Dual Ordinations again in 1989, 1991, twice in 1993, in 1995, 1999 and 2003. Notably, in 1993 she served as principal master of the discipline at the grand ordination ceremony at Luoyang Baimasi monastery 白馬寺, where approximately four hundred *bhikṣuṇī*s were ordained.

47   See Bianchi (2019), Chang (2019, p. 159), Wen (2010), and Xuecheng (1997). For the ordination system in Taiwan, see Li (Forthcoming).

48   The Taiwanese case is presented in Yu-chen Li's studies; particularly see Li (Forthcoming).

49   Tongyuan followed the rule of the "three no's" (*sanbu* 三不), i.e., "not to take disciples, not to have her biography written, and not to write texts promoting her interpretation of Buddhist doctrine" (Péronnet Forthcoming).

50   The last *bhikṣuṇī* ordination before the Cultural Revolution was held in 1957 at Baohuashan 寶華山 (Nanjing).

51   I was not able to trace the translation by Longlian. An English version of Shuyu's text was translated and edited by Bhikṣuṇī Thubten and based on the edition by Jinling Buddhist Scriptures Publishing, Nanjing 2013, which was itself based on a privately published edition of the rite from Tongyuan's collection (Thubten Damcho n.d.).

52   Longlian conferred *śikṣamāṇā*s precepts for the last time in 2005, one year before passing away.

53 On this issue, see Bianchi (2019) and Chang (2019). On the modern "revival movement" of *bhikṣuṇī* ordinations in the different *Vinaya* traditions, see Heirman (2011).

54 Quotation from an interview reported by the journal *Zhongguo fojiao* 中國佛教.

55 The article was translated into Chinese and published in *Fayin* 法音. See Lawei (1982). For the introduction of Dual Ordination in fifth century China, see note 2 above.

56 On the eastern wing of Tiexiang nunnery, a two-story building was built as a housing place for foreign monastics. On these events, see Chang (2019, pp. 159–60), and Qiu (1997, p. 241).

57 Karma Tsomo Lekshe (private communication, December 1997).

58 Sinhalese *bhikkhunī*s were fully ordained at the Sakyadhita conference in Bodh Gaya in 1997; once returned to Śrī Laṅkā, they held the first *bhikkhunī* ordination in centuries at a temple in Dambulla in March 1998 (Ashiwa 2015; see also Huimin 1996). On the revival of the *bhikkhunī* order in the Theravāda tradition, see Anālayo (2013a, 2013b, 2014, 2017). See also Anthony Scott's contributions to this Special Issue, and *bhikkhunī* Dhammadinnā's paper poresented at the conference "Gender Asymmetry in the Different Buddhist Traditions Through the Prism of Nuns' Ordination and Education" (Perugia, 16–17 May 2022).

59 The Karmapa presented a request that Wutaishan Pushou nunnery 普壽寺 *bhikṣuṇī*s bestow ordination on Tibetan nuns in 2015. Pushou nunnery initially responded in a positive way, and an application to be granted permission to ordain Tibetan nuns was presented to the relevant Beijing authorities, but, for 2016, it was not successful. Also Tibetan masters of the Larung gar Five Sciences Buddhist Academy, in Sertar (Easter Sichuan), expressed the wish for Pushousi to cooperate in the ordination of local Tibetan and Chinese nuns. However, Pushou nunnery has yet to receive permission from the government to bestow ordination to Tibetan nuns within PRC borders (private interview with Rurui 如瑞, August 2016). On the establishment of the *bhikṣuṇī* order in the Tibetan tradition, see Heng-Ching (2000), and Roloff (2020). See also Darcie Price-Wallace's contribution to this Special Issue. On Pushou nunnery, see Amandine Péronnet's contribution to this Special Issue.

60 Noteable examples are an interview given in 1995 by Longlian to the journal *Zhongguo fojiao* (quoted by Chang 2019), or the interviews given to Qiu (1997) for the publication of Longlian's biography. Qiu is a female journalist whose aunt was a lifelong friend of Longlian; her book covers Longlian's whole life and includes many interviews and personal anecdotes.

61 There are two documentary films on Longlian by the same title: "The first *bhikṣuṇī* of modern times" (*Dangdai di yi biqiuni* 當代第一比丘尼, Aidaotang 2002, 2009), the shorter one (approx. 30 min), completed in 2002 while Longlian was still alive, and the longer one (approx. 2 h), prepared for her hundrethhundredth birthday commemoration. Both films include videos showing Longlian during rituals and in her everyday life up to her last public appearances.

62 My gratitude to one of the reviewers for bringing this to my attention. For Qiu's (1997) book, see above, note 60.

63 "Even if she has already been accepted and guided by an *upādhyāyinī* as a *śrāmaṇerikā* and *śikṣamāṇā*, she is expected to officially ask a *bhikṣuṇī* to become her *upādhyāyinī* before full ordination" (*T* no. 1428: 924c4–7, quoted by Heirman 1997, p. 51).

64 As Heirman (2001, p. 284) has noted, the *Vinaya* of the Dharmaguptaka "not only present[s] women as a dangerous, weakening factor in the community, but also picture[s] them as beings who are themselves more vulnerable to danger than men are;" as a consequence, the eight *gurudharma*s are also presented "as a bridge or a boat to help women to overcome the dangers of the world".

65 On the notions of Buddhist modernisms, see McMahan (2008, 2012). See also Bianchi (2021b, pp. 11–13 and 18, n. 61).

66 See above, note 4.

67 See Bianchi (2017a). See also Fink (2020, 152 ff).

68 See above, note 64.

69 Digital editions by the Chinese Buddhist Electronic Text Association.

70 This interview was recorded on the occasion of an ordination ceremony held in 1994, and is included in two documentary films about Longlian's life (Aidaotang 2002, 2009). The first and last paragraphs of the interview are included only in the second documentary film (Aidaotang 2009, m. 1,14ff). The rest of the interview is taken from the first documentary film (Aidaotang 2002, m. 11,55–17,50), as the contents of the second have been slightly modified.

71 The four assemblies or orders (*sizhong* 四衆) that make up the Buddhist community, i.e., *bhikṣu*, *bhikṣuṇī*, *upāsaka* and *upāsikā*.

72 This is probably a reference to Ānanda's statement that all Buddhas of the past had four assemblies of disciples. Here Zhongguo ("Middle country") does not refer to China but to the Indian Majjhimadesa (Sk. Madhyadeśa).

73 A three-times *karman*, also known as *(yibai) sanjiemo* (一白)三羯磨 or *sibai jiemo* 白四羯磨 (Sk. *jñapticaturtha karman*), is a formal act occurring during ordinations and repentance ceremonies, entailing a motion and three propositions or responses. For details see Heirman (2002, vol. I, pp. 75–79).

74 This probably refers to a *sīmā* (*jie* 界), a delimited area where formal acts are carried out.

75 This passage is remarkable because it shows Longlian's views in terms of constructing specialization and establishing authority. *Vinaya* itself is a field which does not include the laity. Here Longlian emphasizes that its exclusive nature is even more evident within the ordination system, which excludes non-ordained monastics and non-specialized monastics, and where authority is set on the basis of monastic age and moral value. My gratitude to one of the reviewers for this suggestion.

[76] As noticed by Robert Miller (e-mail personal communication, 29 August 2022), in this interview excerpt *bhikṣuṇī* Longlian seems to be speaking of a nun's *niśraya* ("support" or "dependence"), i.e., the apprenticeship incumbent on all new monks and nuns as conceived in the *Vinaya*.

[77] It is not clear to what event of the *Vinaya* Longlian is referring to. One possibility is that she does not refer to the time of the Buddha in India, but rather to developments in China. There was indeed at least a moment in Chinese history, during the tenth century, when it was established by the imperial government that *bhikṣuṇī* ordinations could only be bestowed by *bhikṣuṇī*s. See above, note 10.

[78] A *benfa ni* is a female candidate who has only gone through the first step of the Dual Ordination (Heng-Ching 2000, p. 515).

[79] The illegitimacy implied here may refer to the fact that *bhikṣu*s bestowing ordination to female candidates without involving *bhikṣuṇī* masters would commit an offernce.

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
