# Peer review of "Reading Equality into Asymmetry: Dual Ordination in the Eyes of Modern Chinese Bhikṣuṇīs"

_religions, doi:10.3390/rel13100919_

Round 1
Reviewer 1 Report
This article makes a crucial contribution to the study of the vinaya revival in modern and contemporary China with a focus on the re-establishment of the dual ordination in mainland China.
The title of this article is “reading equality into asymmetry : Dual ordination in the eyes of modern Chinese bhiksunis.” As I read through the article, I was impressed by the thorough collection of data, but I don’t see a direct discussion on gender equality and asymmetry. I think the author can make this statement more explicitly. One crucial source that has been mentioned but not mainly used by the author is the biography of Longlian as authored by Qiu Shanshan. In this book, Longlian expressed her view on gender equality and her aspirations for re-establish the female Buddhist order. If the author can make more reference to this work, the argument made by the author on P. 18 in the paragraph before the conclusion, the author can make a stronger case for the argument. A second point is that the author includes the effort of both male and female Buddhists, especially in section 2. Thus, it seems that the title does not show the multi-perspective-ness of the discussion. Should the title be modified to show such a diversity of perspectives?
Some translation issues can be found in section 3. The author needs to keep in mind that Longlian sometimes speaks the colloquial Sichuan dialect. That is why there are some repetitions and local expressions. For instance, in the first long quote on P. 14, the author omits a lot of sentences towards the end of this excerpt. The following long quote on P. 15, “光是十個比丘通過還不行”, here 還不行 is a local expression in Sichuan, wchich means “not enough” rather than “not good” as translated by the author. I would recommend the author to double check the translated passages.
Besides, the author needs to keep consistency of translation. For instance, 文殊院 is sometimes translated as Wenshu temple, sometimes as Wenshu monastery, and sometimes transliterated as Wenshuyuan. The author might want to keep it consistent.
Author Response
I am very grateful to both reviewers for their useful feedback, criticism and suggestions. I have tried to address many of the issued raised by both. See below an explanation of the changes and additions I made accordingly, distinguished between contents and form. At the end of this cover letter, I explain separately to each reviewer why I decided not to follow a few of their suggestions. Of course, I am ready to revise these decisions, in case they deem it necessary.
FORM
I addressed some English language issues (rephrased some sentences, changed some terms, corrected misspellings and mistakes), and also corrected some typos in the Sanskrit terminology and Chinese pinyin transcriptions.
As suggested by Reviewer 1, for consistency I uniformized names of some monasteries (e.g. Wenshuyuan) and other recurring terms.
I also revised the translations in section 2 and made some corrections and changes, as suggested by Reviewer 1 (whom I thanks for pointing this out to me).
CONTENTS
As for contents, I tried to make my arguments clearer, by adding some connecting sentences and summarizing passages. This point was pointed out by both Reviewers (with different degrees), and I hope my changes are acceptable to both. The major additions are highlighted in yellow in the revised article.
Changes and additions include:
I added an explanation of what I mean by asymmetry and discuss the notions of asymmetry/equality as they are understood in my article, as requested by Reviewer 1. I also included a reference to the notion of orthodoxy as a major thread of the article.
Among the major changes, as suggested by Reviewer 1, I referenced Qiu Shanshan’s volume in regard to Dual Ordination and gender equality. I am very indebted for this, as the Reviewer notices, I have referenced Qiu’s book in my essay, and I have used it a lot in my previous study on Longlian also in regard to the Dual Ordination issue, but for some reason I did not do so this time. An acknowledgment is to be found in footnote 77. I noticed that Reviewer 1 agrees to sign his/her open review; therefore, I wonder, if it would not be possible to quote her/his name in the footnote. Please let me know.
NOTE FOR REVIEWER 1
It is true that the title does not show the involvement of the male Buddhists and thus the multi-perspective-ness of the discussions described in section 2. Yet, in my article I wanted to emphasize the role of the bhikṣuṇīs themselves, regardless by the fact that background and impulse were mainly provided by their male masters. For this reason, I chose not to modify my title. Hope this is acceptable to you.
Reviewer 2 Report
1. The article attempts to describe Longlian’s and Tongyuan’s promotion of the twofold process of ordination, but the arguments present these two nuns as the primary and exclusive factors that China reestablished twofold ordination, which should be revised. In fact, these two nuns were only part of the factors. The author should not overstate the two nuns’ roles in Chinese history:
Abstract “The rule…. due mainly to the efforts of bhikṣuṇīs Longlian 隆蓮 11 (1909–2006) and Tongyuan 通願 (1913–1991).”
P. 4 “This was mainly due to the efforts of two prominent bhikṣuṇīs”
2. The article would be more convincing if the author systematically depicted the broader background, including governmental policy, BAC, Taiwan’s influence (p. 13), and Longlian’s political stance (author only mentions in note 47).
3. Session 3.2—the author should consider analyzing the block quotes. For example, “要在僧團當中選出十位地位特別高的,道高德重,戒臘須彌的人,來開這個特別 的審查會” “這個會就是說其他人不能參加” “只有一個地方受都是不合法的”—these statements illustrate Longlian’s views on constructing specialization and establishing authority.
4. Although the author mentions the “notion of orthodoxy” in the conclusion, this theme does not well developed in the discussion. The concern about “orthodoxy” can be the main thread to linking the arguments in the article.
5. The article mainly describes Longlian’s views, but it has fewer sources related to Tongyuan.
6. Language: the term bhikṣuṇīs (Sanskrit) does not exactly use in China. Instead, the Chinese pronunciation 比丘尼 biqiuni sounds similar to Pali, bhikkhunī. For readers’ convenience, it would be more suitable to call them nuns.
Author Response
I am very grateful to both reviewers for their useful feedback, criticism and suggestions. I have tried to address many of the issued raised by both. See below an explanation of the changes and additions I made accordingly, distinguished between contents and form. At the end of this cover letter, I explain separately to each reviewer why I decided not to follow a few of their suggestions. Of course, I am ready to revise these decisions, in case they deem it necessary.
FORM
I addressed some English language issues (rephrased some sentences, changed some terms, corrected misspellings and mistakes), and also corrected some typos in the Sanskrit terminology and Chinese pinyin transcriptions.
CONTENTS
As for contents, I tried to make my arguments clearer, by adding some connecting sentences and summarizing passages. This point was pointed out by both Reviewers (with different degrees), and I hope my changes are acceptable to both. The major additions are highlighted in yellow in the revised article.
Changes and additions include:
I added an explanation of what I mean by asymmetry and discuss the notions of asymmetry/equality as they are understood in my article. I also included a reference to the notion of orthodoxy as a major thread of the article, as mentioned by Reviewer 2.
As suggested by Reviewer 2, I clarified that the role of Longlian and Tongyuan for the establishment of Dual Ordinations is prominent in PRC’s 1980s, while during the Republic the topic was already addressed by others and later it was implemented in Taiwan by other masters, both male and female. It was not my intention “to overstate the two nuns’ role in Chinese history”, but rather to present their role in that particular moment and space. Thanks for suggesting this crucial revision!
As for Tongyuan, as far as I know, there is not much more on her views about Dual Ordination. As she refrained to write, her ideas only come across her actions and other persons’ descriptions. However, as requested by Reviewer 2, I added some additional sentences, and explained the reason for this lack of information.
Notes have been added in the translation of Longlian’s interview, in order to give more context and analysis of its contents, as suggested by Reviewer 2.
NOTES FOR REVIEWER 2
- The “broader background”, including governmental policy, BAC, and Longlian’s political stance are already discussed in another article of mine. This is why I only mentioned these issues in the footnote, only summarizing the main points in the main text. I hope this is acceptable to you.
- I employed bhikṣuṇī (instead of bhikkhuni) for consistency, as all terms in my article referring to Chinese Buddhism are in Sanskrit (I employ Pali only in reference to Theravada ‘nuns’). The reason why I did not employ “nun” is that in the Special Issue the term “nun” refers to all the different cases of Buddhist female monastics throughout Asia; by using bhikṣuṇī I distinguish the full ordained Chinese monastics from the other forms of female monastic commitment.
Round 2
Reviewer 2 Report
The article has sufficient revision and the overall arguments are interconnected.